# Effects of Extracts Containing Metabolites of Different Cyanobacteria from an Ambient Spring (Central Europe) on Zooplankters *Daphnia magna* and Duckweed *Spirodela polyrhiza*

**Magdalena Toporowska** [1,*], **Beata Ferencz** [1] and **Jarosław Dawidek** [2]

1   Department of Hydrobiology and Protection of Ecosystems, University of Life Sciences in Lublin, 13 Akademicka St, 20-950 Lublin, Poland
2   Department of Hydrology and Climatology, Maria Curie-Skłodowska University, Kraśnicka 2cd, 20-718 Lublin, Poland
*   Correspondence: magdalena.toporowska@up.lublin.pl; Tel.: +48-814-610-061 (ext. 309)

**Abstract:** Toxic cyanobacterial blooms are a well-known problem in eutrophic water bodies, but cyanobacterial toxicity in ambient springs is unexplored. Therefore, we studied the toxin content and effects of two extracts obtained from epilithic and benthic microbial mats containing different algae and filamentous cyanobacteria, *Phormidium breve* and *Oscillatoria limosa*, respectively, on fresh biomass, number of roots, and pigment content in duckweed *Spirodela polyrhiza* and on survivorship of *Daphnia magna* (Cladocera). Mat samples, used to prepare extracts for bioassays, were collected in the anthropogenically transformed limnocrenic Górecko spring, located (50°31′08″ N and 22°57′10″ E) in the Roztocze region (Eastern Poland). It drains an abundant aquifer built with Cretaceous sediments. Neither anatoxin-a (ANTX) nor microcystins (MCs) were detected in the extracts using HPLC techniques; however, negative effects of the extracts on tested organisms were observed. The *Phormidium* extract contained concentrations of cyanobacterial metabolites a few times higher than the *Oscillatoria* extract. In general, both extracts affected plants in a dose-dependent manner; however, the *Oscillatoria* extract influenced pigment production without a clear trend. The highest concentrations of *Phormidium* extract ($p < 0.05$) significantly decreased the number of roots and the content of chlorophylls and carotenoids in *S. polyrhiza*. The *Oscillatoria* extract caused a statistically significant ($p < 0.05$) decrease in biomass and number of roots; however, it generally positively influenced the production of pigments. The extract containing *O. limosa* metabolites was more toxic to *D. magna* than the extract containing higher amounts of metabolites of *P. breve*. Cyanobacteria inhabiting temperate springs may negatively affect hydrobionts by producing secondary metabolites other than ANTX and MCs; however, the contribution of algae cannot be excluded. Extensive research on cyanobacteria in springs, their metabolites, and their effects on living organisms should be conducted.

**Keywords:** toxicity; limnocrens; aquatic macrophytes; zooplankton; microcystins; anatoxin-a

## 1. Introduction

Cyanobacteria are heterogenic, photosynthetic organisms that occur commonly in different aquatic and terrestrial environments [1]. The massive development of cyanobacteria, as well as the production of toxins and other biologically active cyanobacterial secondary metabolites occurring worldwide in various water bodies, have been the subject of intensive studies in recent decades [2–6]. In contrast, less attention has been paid to cyanobacteria inhabiting ambient springs (characterized by a temperature approaching the mean annual air temperature (MAAT) in the drainage basin [7]), in which researchers have focused mostly on cyanobacterial community structure [8–11]. Springs are extremely diverse ecotones characterized by a remarkable variety of environmental parameters (from highly

shaded to UV-exposed, from extremely soft to carbonate-precipitating water, from permanent discharge to intermittent flow, from still water to strong currents, etc.), which can be classified in accordance with different criteria derived from geology, water temperature, hydrology, hydrochemistry, or human use [7]. Therefore, they are very important habitats for biodiversity conservation [12]. Moreover, springs are often one of the last high-integrity, oligotrophic freshwater ecosystems in populated areas and are subjected to anthropogenic impact and climate change [12,13]. Despite their variability, springs were ignored for a long time by hydrobiologists [12] including studies on cyanobacteria and algae other than diatoms. Few studies published over the last 20 years [7–10] have shown that in springs, cyanobacteria may constitute one of the richest taxonomies and most numerous groups of microbiota subject to seasonal dynamics [10].

In contrast to planktonic cyanobacteria, the potential toxin production and toxicity of cyanobacteria inhabiting springs are studied extremely rarely [14,15]. The knowledge of planktonic cyanobacteria and their ability to produce different biologically active and often toxic metabolites is broad [3,4,6]. For instance, some planktonic cyanobacteria can produce microcystins (MCs), which exhibit hepatotoxicity, nephrotoxicity, and neurotoxicity, and can cause cardiovascular disease, immunomodulation, endocrine disruption, reproductive and developmental toxicity [16], and other biologically active, mostly non-ribosomal oligopeptides such as aeruginosins, cyanopeptolins, anabaenopeptins, microviridins [3,17], as well as neurotoxic anatoxins (ANTXs) or saxitoxins (STXs) and many others substances that may negatively affect aquatic organisms [6]. Findings on cyanotoxins in extreme ecosystems such as ambient springs or streams are rare [14,18–21]. For example, a benthic cyanobacterium, *Iningainema pulvinus* gen nov., sp. nov., that produced nodularin, was recently isolated from a freshwater ambient spring in tropical, northeastern Australia [14]. Two extracts from the strain contained 796 and 1096 μg of nodularin/g dry weight. Cantoral Uriza and co-authors [15] recently showed the production of MCs by two of four cyanobacterial species isolated from freshwater springs in Mediterranean marshes. More often, studies demonstrate the production of toxins by benthic cyanobacteria inhabiting streams [18–20]. In many regions, including Roztocze, streams originate from the spring's area. Fetscher and co-authors [20] demonstrated the production of MCs and ANTX in California streams, whereas Aboal and co-workers [19] demonstrated that cyanotoxin production is common in calcareous Mediterranean streams of a high-ecological-integrity. Since several of the species for which MC production was shown (*Rivularia* spp., *Tolypothrix distorta*, *Schizothrix fasciculata*, *Oscillatoria* spp., and *Phormidium* spp.) occur also in springs [9,10,19], it is very probable that cyanotoxins and/or other biologically active cyanobacterial secondary metabolites may be widely produced by cyanobacteria inhabiting ambient springs as well, especially since such a production has been confirmed for frequently studied hot springs [22,23].

The toxicity of cyanobacteria to living organisms was demonstrated mostly in studies on planktonic species, and the negative effects of various cyanobacterial metabolites on aquatic organisms, ecosystems, and human health have been observed worldwide [3,6,17,24]. Moreover, Miller and co-authors [25] provided evidence suggesting the land–sea flow of MCs with trophic transfer through marine invertebrates as the most probable route of exposure as they confirmed deaths of sea otters from MC intoxication. This finding illustrates cyanotoxins' and/or other cyanobacterial bioactive compounds' negative effects on ecosystems, even far downstream from their origin, because of fluvial transport. Despite the growing evidence that benthic cyanobacteria are an important source of cyanotoxins and T&O (taste and odor) compounds in waters globally [26], this phenomenon is almost unexplored in ambient springs, which usually feed river systems with water. In springs and other aquatic ecosystems, benthic cyanobacteria can grow on different substrates including sediment and stones. They can also detach and colonize macrophytes. Benthic cyanobacteria can also create floating mats [8]. Therefore, their role, particularly in limnocrenic springs (springs that form pools), may be very important and needs to be studied.

The species *Spirodela polyrhiza* (giant duckweed) is a cosmopolite representative of the Lemnoideae subfamily in standardized ecotoxicological test procedures [27]. *S. polyrhiza* is a good model for measuring stress, and also points to turion and root importance. It is widely applied as a model organism in plant physiology, ecotoxicology, and bioremediation studies [28,29]. Crustacean *Daphnia magna* (Cladocera) also is used worldwide as a model organism, and the role of bioassays using both species in ecotoxicological studies is growing [28–31]. Therefore, these species were chosen for the present study. The aim of this paper was to study the production of the most common cyanotoxins, hepatotoxins (microcystins), and neurotoxins (anatoxin-a) by cyanobacteria present in benthic microbial mats which developed in the anthropogenically transformed Górecko spring. We also studied the effects of the crude extracts obtained from the mat samples on zooplankters *D. magna* (survivorship) and the macrophyte *S. polyrhiza* (fresh biomass, number of roots, and production of chlorophylls and carotenoids). We analyzed the production of pigments by the macrophyte, as they play a crucial role in photosynthetic activity. Chlorophyll-a (Chl-a) is the main photosynthetic pigment, whereas Chl-b is a ubiquitous accessory plant pigment, and the ratio of Chl-a to Chl-b controls the absorbed light intensity [32]. We hypothesized that the extracts negatively affect *D. magna* survivorship and development of *S. polyrhiza* due to the presence of MCs and/or ANTX or other biologically active secondary metabolites.

## 2. Materials and Methods

### 2.1. Study Area, Field Study, and Physical-Chemical Analysis of Spring Water

According to the physiographic division of Poland, the Górecko spring (50°31′08″ N and 22°57′10″ E) is located in the middle Roztocze in the edge zone of the Biłgoraj Plain mesoregion (eastern Poland, Figure 1). The spring is located 243.8 m above sea level. The area is built with Cretaceous geodes, marls, and opokas. They are covered with a layer of Quaternary sediments: Pleistocene river sands from accumulation terraces, as well as Holocene sands and river fen soils. The under-slope Górecko spring is located in the Szum River catchment area, the right-bank tributary of the Tanew River. It is a gravity spring and drains an aquifer built with pored and fissured Cretaceous rocks. The human activity impact on the spring is evident. The niche, located in an anthropogenically transformed area (Figure 1), is enclosed with a cement–stone wall that has the regular shape of a 10 m wide and 8 m long rectangle. The area of the spring niche is 79 m$^2$ with a maximum depth of 40 cm.

The average efficiency of the spring was measured using a Valeport 801 Electromagnetic Flow Meter (Valeport Limited, Totnes, UK). The basic physical parameters of the spring water, including water temperature, conductivity, pH, and oxygen concentration, were measured in situ using a multiparameter meter (YSI ProDSS, YSI, Yellow Springs, OH, USA). $NH_4$, $NO_3$, $PO_4$, and total phosphorus (TP) concentrations in the water were analyzed using an HPLC-photodiode array detection system (Shimadzu, Kyoto, Japan). To measure anion concentrations, water was filtered through cellulose–acetate membranes (0.45 mm pore size), whereas cation concentrations were analyzed in acetified filtered water (*p* ~2).

### 2.2. Collection and Microscopic Analyses of Microbial Mats

Collections were made in August 2020 at two sites, located in different parts of the Górecko spring, where biofilms were visually detectable as green or yellow-green microbial mats on stones and sediments. Samples from sediments were collected using a syringe and from stones using a scalpel. Then the samples were placed in containers with ice and transported to a laboratory. The taxonomical identification of phycoflora, including cyanobacteria, was carried out using a light microscope (Zeiss Primo Star, Zeiss, Jena, Germany) and professional manuals [33–35]. Diatoms were identified, mostly up to the genus level, according to Cox [34]. The abundance and biomass (quantitative composition) of cyanobacteria and algae were analyzed using light microscopy in a plankton Sedgewick–

Rafter chamber (Paul Marienfeld GmbH & Co. KG, Lauda-Königshofen, Germany) after dilution of samples (1:20). Biomass of cyanobacteria and algae was calculated according to Hillebrand and co-authors [36]. Two different phycoflora samples for bioassays were frozen (–20 °C) until extraction (for 10 min, 50 W, ultrasonic homogenizer Sonopuls, Bandelin, Berlin, Germany) of crude extracts, cyanotoxin analysis, and biotests.

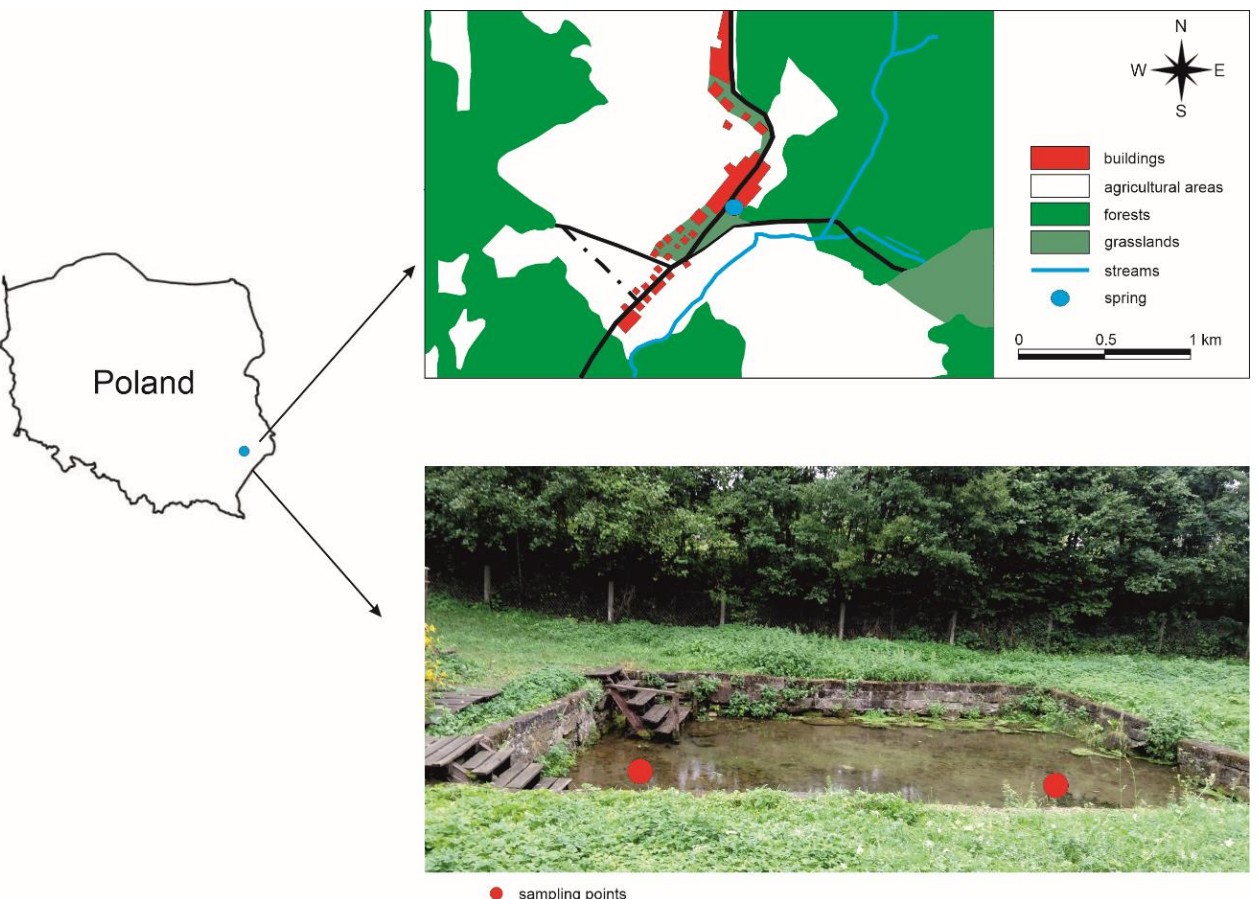

**Figure 1.** Location of Górecko spring and sampling points.

### 2.3. Extraction of Phycoflora Biomass

For the determination and identification of intracellular microcystins (MCs) and anatoxin-a (ANTX) in samples of phycoflora mats, 2 mL of biomass was filtered through Whatman GF/C filters, which was frozen until the day of cyanotoxin extraction and analysis. Extracts of biomasses collected on filters were prepared using ultrasonication (2 times for 10 and 5 min, respectively; 50 W, ultrasonic homogenizer Sonoplus, Bandelin, Germany) in 75% (*v/v*) methanol (gradient-grade, Merck, Frankfurt, Germany) containing 0.002 M HCl. Supernatants were collected after centrifugation (14,000 rpm for 10 min, 17 °C) and frozen (–20 °C) until cyanotoxin analysis. Samples of mats were also used to prepare crude extracts for the determination of MCs and ANTX, and for experimentation. The biomasses were sonicated (for 20 min, 50 W), and supernatants were collected after centrifugation (14,000 rpm for 10 min at 17 °C). Toxins were analyzed in the extracts before experimentation.

### 2.4. HPLC-PDA Analysis of Microcystins

For MC detection and identification, the HPLC-photodiode array detection system (Shimadzu, Kyoto, Japan) was used. The UV detection range was 200–300 nm. Microcystin-RR, MC-YR, [D-Asp³]MC-LR, MC-LR, MC-LW, MC-LA, MC-LY, MC-WR, and MC-LF (Enzo Life Science, Lausen, Switzerland) were used as the standards. Separation of extract

components was performed with the aid of a Purosphere column (125 × 3 mm, dp 5 μm, Merck, Darmstadt, Germany), with the mobile phase composed of acetonitrile (Merck, Burlington, VT, USA) and gradient-grade water for HPLC that was acidified with 0.05% trifluoroacetic acid (gradient 30–100%). The flow rate was 0.7 mL/min.

*2.5. HPLC-FLD Analysis of Anatoxin-a*

ANTX in extracts was analyzed using liquid chromatography (HPLC) with fluorescence detection (Shimadzu, Kyoto, Japan), according to James and co-authors [37]. A percentage of 10% NBD-F (4-fluoro-7-nitrobenzofuran; Fluka) was used for ANTX derivatization. The detector parameters were as follows: the excitation wavelength was 470 nm, whereas the emission wavelength was 530 nm. Separation of extracts was obtained using a Purosphere column (125 × 3 mm, dp 5 μm Merck) and TFA (0.05%) acidified acetonitrile and gradient-grade water at a flow rate of 0.6 mL/min. Standard ANTX (Enzo Life Sciences, Lausen, Switzerland) was used for cyanotoxin identification.

*2.6. Toxicity Bioassays*
2.6.1. *S. polyrhiza* Bioassay

The toxicity bioassays on *S. polyrhiza* (Duckweed Toxkit F, MicroBio Tests Inc., Gent, Belgium) were performed in six replicates in cups (2 mL in volume) of multi-well plates in the standard medium. Each 72 h-old plant was placed into a cup containing five dilutions of the extract obtained from the microbial mats containing cyanobacteria. Crude extracts used in the bioassays were diluted, and the final concentration ranges of cyanobacterial biomass were 2.5-fold higher in the *Phormidium* extract than in the *Oscillatoria* extract. They reached 25–400 mg/L in the case of the *Phormidium* extract and 10–160 mg/L for the *Oscillatoria* extract while maintaining similar concentrations of the total phycoflora biomass (from 41 to 656 mg/L for the *Phormidium* extract and from 47 to 748 mg/L for the *Oscillatoria* extract). Controls (in six replicates) were also set. The young plants were incubated for the following 72 h at 25 °C with continuous illumination (6000 lux) according to the producer's (MicroBio Tests Inc., Gent, Belgium) protocol.

2.6.2. *Analyses of Growth and Development Parameters*

After the 72 h exposure, the total biomass (expressed as fresh weight—FW to an accuracy of 0.1 mg) of each individual was measured after gentle drying with filter paper and weighing, the number of roots of each plant was counted directly, the analysis of pigments Chlorophyll-a (Chl-a), Chl-b, and carotenoids (Car) in plants from replicates no. 1, 3, and 5 were carried out, and the ratio of Chl-a to Chl-b was calculated.

2.6.3. *Analysis of Pigments*

Plants from replicates no. 1, 3, and 5 were grinded with the aid of a glass spatula and then homogenized in 1.5 mL of 90% ethanol for 5 min at 75 °C. Then the homogenate was centrifuged at 14,000× *g* for 10 min. The Chl-a, Chl-b, and carotenoid (Car) absorbance in the extracts was measured at 470, 649, and 665 nm with the spectrophotometer (SPECORD 40, Analytik Jena GmbH, Jena, Germany).

2.6.4. *Acute Toxicity Bioassays on Daphnia Magna*

The toxicity of the extracts towards the cladoceran *D. magna* (Daphtoxkit FTM magna, MicroBio Tests Inc., Gent, Belgium) was evaluated in 48 h bioassays. In each of the five replicates, five specimens of *D. magna* were exposed in a 0.3 mL standard medium. The bioassays were performed at a temperature of 20 °C, in darkness, according to the producer's (MicroBio Tests Inc., Gent, Belgium) protocols using a range of extract dilutions in a standard medium. The biomass of the extracted cyanobacteria in the extracts was ca. 3.6-fold higher in *Phormidium* extract than in *Oscillatoria* extract. They ranged from 25 to 400 mg/L in treatments containing *Phormidium* extract and from 7 to 112 mg/L in treatments with *Oscillatoria* extract while maintaining similar concentrations of the total

phycoflora biomass (from 41 to 656 mg/L for *Phormidium* extract and from 31 to 496 mg/L for *Oscillatoria* extract). Five dilutions of extracts were tested. The test endpoint was the death of the daphnids. As controls, daphnids were incubated in the standard medium.

*2.7. Data Analysis*

　　All of the data that characterize microbial mats and extracts were expressed as mean values ($n$ = 2–4) $\pm$ standard deviation (SD), whereas all of the data obtained in the bioassays were shown as mean values ($n$ = 3–6) $\pm$ standard error (SE). Normality distribution (Kolmogorov–Smirnov test) and homogeneity of variance (Levene's test) were tested for the data obtained from bioassays. Significant differences among treatments were evaluated using a one-factor analysis of variance (ANOVA). A pairwise comparison of means was done using the Tukey test ($p < 0.05$). The statistical tests were carried out using the Statsoft Statistica package v. 10 for Windows (a significance level of $p < 0.05$). Toxicity data obtained in the tests on *D. magna* were presented as the percentage (%) of the organisms' survivorship compared to the controls. For the determination of 48 h $LC_{50}$ (the acute toxicity parameter), probit analysis was used. Values of $LC_{50}$ with non-overlapping 95% intervals were regarded as significantly different.

## 3. Results

*3.1. Hydrological and Physicochemical Characteristics of Górecko Spring*

　　The average discharge of the anthropogenically transformed Górecko spring in August 2020 was 6 L/s. The water temperature varied slightly and has not exceeded 10 °C (from 9.8 to 9.9 °C), which classifies the spring as ambient. Both pH value (7.4) and oxygen concentration (3.38 mg/L) resulted from the impact of the closed hydrogeochemical system in the water in the spring's niche [38]. Although the spring's niche was affected by human activity, the water type (determined by ionic composition) does not evolve from $HCO_3Ca$. The percentage share of $Ca^{2+}$, $Mg^{2+}$, $Na^+$, $K^+$, and $Sr^{2+}$ (in mval/L) in the sum of cations was 91.18, 4.57, 2.95, 1.15, and 0.14%, respectively. The share of $HCO_3^-$, $SO_4^2$, $NO_3^-$, $Chl^-$, $F^-$, and $PO_4^{3-}$ in the sum of anions was 81.08, 8.49, 7.80, 2.15, 0.27, and 0.21%, respectively. The value of electrical conductivity (488 μS/cm) is within the typical range (average for the region) of 483 μS/cm, according to Chmiel [39], of Roztocze's springs. Only slightly elevated concentrations of anthropogenic $K^+$ and $SO_4^{2+}$ (2.6 and 25.2 mg/L, respectively) and nutrients (29.9 mg $NO_3$/L and 0.41 mg $PO_4$/L), compared to other of Roztocze's springs [40], were observed.

*3.2. Characteristics of Microbial Mats and Extracts*

　　Mat samples used to prepare extracts for bioassays originated from two sampling points located in the Górecko spring (Figure 1) and differed in biomass and composition of phycoflora (Table 1). One sample was composed of cyanobacteria predominated by *Phormidium breve* and Bacillariophyceae and was used to prepare *Phormidium* extract. The second sample was composed of Bacillariophyceae, filamentous Ulvophyceae, and cyanobacteria predominated by *Oscillatoria limosa* and was used to prepare *Oscillatoria* extract. The total biomass of phycoflora in the mat with *P. breve* was almost two-fold higher (4634.0 mg FW/L) than the total phycoflora biomass in the mat containing *O. limosa* (2625.3 mg FW/L), whereas the biomass of cyanobacteria was almost five-fold higher in the mat with *P. breve* than the biomass of cyanobacteria in the mat with *O. limosa*. In total, five cyanobacterial taxa were found in benthic mats in the Górecko spring. In both mats, three accompanying cyanobacterial taxa occurred. Their biomass did not exceed 5% of the total cyanobacterial biomass. HPLC analysis of intracellular cyanotoxins revealed neither MCs nor ANTX in microbial mats containing cyanobacteria (Table 1).

**Table 1.** Characteristics of microbial mats used to prepare crude extracts for bioassays. Data are expressed as means $\pm$ SD ($n = 4$).

| Parameters | *Phormidium* Mat/Extract | *Oscillatoria* Mat/Extract |
|---|---|---|
| The total phycoflora biomass (mg FW/L) | 4634.0 $\pm$ 602.4 | 2625.3 $\pm$ 183.8 |
| The cyanobacterial biomass (mg FW/L) | 2799.5 $\pm$ 370.7 | 583.5 $\pm$ 70.0 |
| Cyanobacterial taxa and their contribution to the total cyanobacterial biomass (%) | *Phormidium breve* (95.8%) *Leptolyngbya* sp. 1 (3.3) *Oscillatoria limosa* (0.8) *Leptolyngbya* sp. 2 (0.1) | *Oscillatoria limosa* (95.1%) *Phormidium breve* (4.4) *Leptolyngbya* sp. (0.4) *Pseudanabaena minima* (0.1) |
| Biomass of eukaryotic algae (mg FW/L): | | |
| Bacillariophyceae | 1834.5 $\pm$ 302.7 | 1033.6 $\pm$ 98.2 |
| Chlorophyceae | n.f. | 0.7 $\pm$ 0.03 |
| Filamentous Ulvophyceae | n.f. | 1007.5 $\pm$ 6.5 |
| Microcystins (µg/L) | n.d. | n.d. |
| Anatoxin-a (µg/L) | n.d. | n.d. |

Note: n.f.—not found; n.d.—not detected.

### 3.3. Effects of the Extracts on the Growth and Development of S. Polyrhiza

Figures 2 and 3 show that after 72 h of exposure, extracts containing similar total biomass of phycoflora but different biomass of cyanobacteria affected the biomass (FW) and root system of young individual *S. polyrhiza*. After exposure to *Phormidium* extract and *Oscillatoria* extract, the average biomass of *S. polyrhiza* decreased by 43% and 47%, respectively (from 5.42 in the controls to 3.07 mg of FW in the treatment with the highest extract concentration used, and from 5.78 in the controls to 3.08 mg of FW in the penultimate concentration used, respectively; Figure 2, Table 2). The average number of roots (Figure 3) incubated with *Phormidium* extract decreased by 70% in comparison with the controls (from 6.5 to 2), whereas in the treatments with *Oscillatoria* extract, the parameter decreased by 60% (from 7.4 to 3.1). The strongest, most negative, and statistically significant ($p < 0.05$) effect on fresh plant biomass (Figure 2) and number of roots (Figure 3) was observed at the highest extract concentrations used, particularly in the case of *Phormidium* extract (Figures 2a and 3a). The most negative effect of *Oscillatoria* extract on *S. polyrhiza* both in the case of fresh biomass (Figure 2b) and the number of roots (Figure 3b) was observed at the concentration of phycoflora biomass equal to 374 mg/L. At two-fold higher concentrations of *Oscillatoria* extract, both the biomass and number of roots increased; however, the differences between both treatments were not statistically significant.

**Table 2.** Comparison of toxic effects of two the highest concentrations of the extracts obtained from similar phycoflora biomass and different cyanobacterial biomasses on *S. polyrhiza,* showing the importance and dependence of the effects observed on the concentration of cyanobacterial biomass.

| Extract | Phycoflora Biomass (mg/L) | Cyanobacterial Biomass (mg/L) | Growth Inhibition (%) | |
|---|---|---|---|---|
| | | | *S. polyrhiza* Biomass | Number of Roots |
| *Phormidium* | 328; 656 | 200; 400 | 24 $\pm$ 4.5; 43 $\pm$ 4.9 | 51 $\pm$ 2.6; 69 $\pm$ 5.6 |
| *Oscillatoria* | 374; 748 | 83; 166 | 47 $\pm$ 3.2; 23 $\pm$ 6.7 | 58 $\pm$ 4.9; 37 $\pm$ 6.5 |

Note: Results for growth inhibition are expressed as means $\pm$ SE ($n = 6$). *S. polyrhiza* biomass and number of roots in controls were set as 100%.

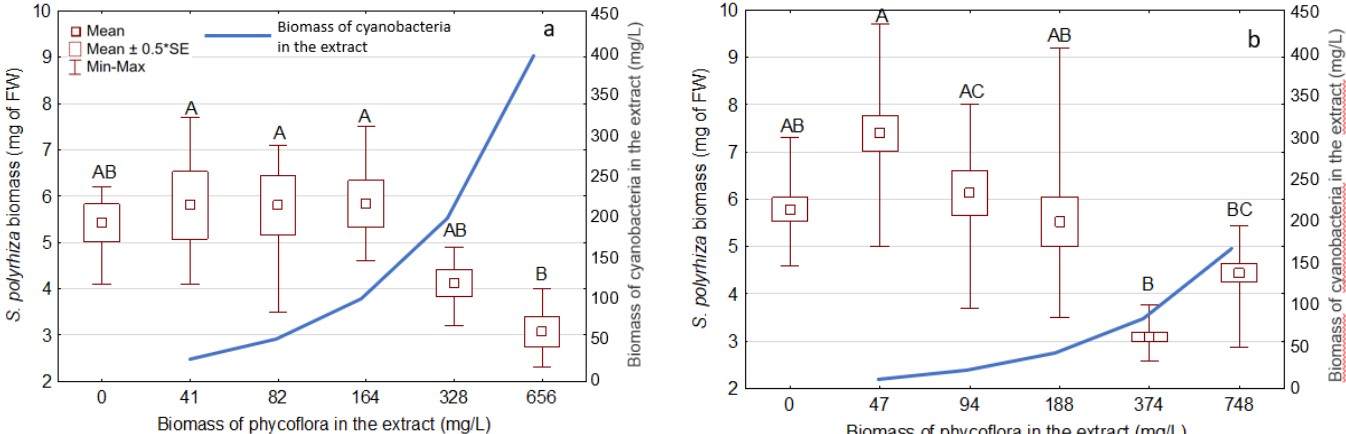

**Figure 2.** Effect of *Phormidium* extract (**a**) and *Oscillatoria* extract (**b**) on the fresh biomass of *S. polyrhiza* after 72 h exposure (means ± SE, *n* = 6, 0 = controls). Different uppercase letters (A, B, C) indicate statistically significant differences between treatments (ANOVA, Tukey's test, *p* < 0.05).

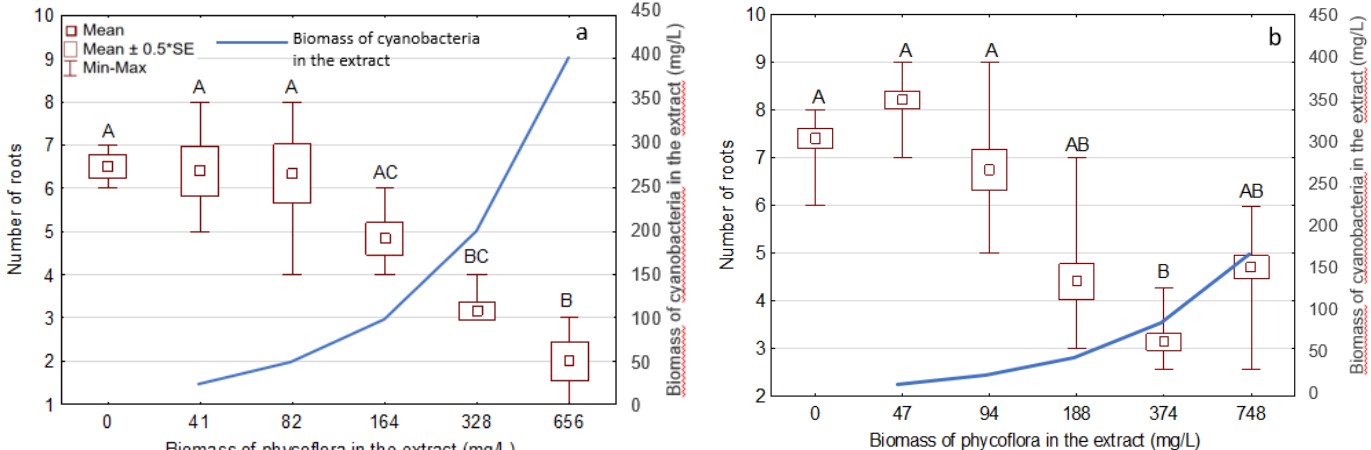

**Figure 3.** Effect of *Phormidium* extract (**a**) and *Oscillatoria* extract (**b**) on the number of roots of *S. polyrhiza* after 72 h exposure (means ± SE, *n* = 6, 0 = controls). Different uppercase letters (A, B, C) indicate statistically significant differences between treatments (ANOVA, Tukey's test, *p* < 0.05).

A comparison of the toxic effects (Table 2) caused by the extracts containing different concentrations of cyanobacterial biomass, albeit obtained from similar phycoflora biomasses, suggests a prominent contribution of the cyanobacterial metabolites to the toxicity observed in *S. polyrhiza*. The inhibition of development of root systems (by 37–69%) was stronger than the inhibition of the total plant biomass (23–47%).

Generally, the exposure of *S. polyrhiza* to different extracts containing cyanobacterial metabolites resulted in varied reactions in the production of pigments by the plants (Figure 4). The content of Chl-a *in S. polyrhiza* exposed to the *Phormidium* extract (Figure 4a) decreased in a dose-dependent manner, whereas in plants exposed to *Oscillatoria* extract, it changed without a clear trend (Figure 4b). A similar phenomenon was observed in the case of the production of Chl-b (Figure 4c,d) and carotenoids (Car) (Figure 4e,f). The *Phormidium* extract caused a significantly important (*p* < 0.05) decrease in the content of chlorophylls in plants under the two highest concentrations used (328 and 656 mg/L). On the contrary, the highest concentration of the *Oscillatoria* extract induced a statistically important increase in the content of both Chl-a (Figure 4b) and Chl-b (Figure 4d) in macrophytes. Both extracts caused changes in the Chl-a/Chl-b ratio (Figure 4g,h). In both bioassays, it was a ca. 1.2-fold decrease. In the bioassay with *Phormidium* extract (Figure 4g), the ratio decreased from 2.68 in the controls to 2.22 at the highest extract concentration used, whereas in the bioassays with *Oscillatoria* extract (Figure 4h), the ratio decreased from 2.83 in the controls

to 2.26 in the one before the last experimental variant. The highest concentrations caused a decrease in the ratio that was significant under the two highest concentrations of the *Phormidium* extract (Figure 4g) and at the concentration of 374 mg/L of *Oscillatoria* extract (Figure 4h).

### 3.4. Effects of the Extracts on D. magna Survivorship

The 48 h exposure of the cladocerans *D. magna* to five concentrations of the extracts cause a strong decrease in zooplankters survivorship at the highest concentrations used, with significant differences in comparison with controls (Figure 5). The survivorship of daphnids was reduced by 70% after exposure to both extracts, compared with the controls (Figure 5). However, comparing the biomass concentrations in the extracts, the *Phormidium* extract showed slightly lower toxicity to *D. magna* than the *Oscillatoria* extract (Figure 5, Table 3). Since the biomass of cyanobacteria present in the extracts was 3.6 times higher in the *Phormidium* extract than in the *Oscillatoria* extract, the 48 h $LC_{50}$ value based on cyanobacterial biomass was over two-fold lower for *Oscillatoria* extract (81 mg/L) than for *Phormidium* extract (181 mg/L) (Table 3).

**Table 3.** Toxicity (48 h $LC_{50}$) of extracts to *D. magna*, determined on the basis of the concentration of the total phycoflora biomass and cyanobacterial biomass present in the extracts (mean $\pm$ SD; $n = 4$).

| Extract | 48 h $LC_{50}$ Expressed as mg/L | |
|---|---|---|
| | **Total Phycoflora Biomass** | **Cyanobacterial Biomass** |
| *Phormidium* | 381 $\pm$ 82 | 181 $\pm$ 61 |
| *Oscillatoria* | 369 $\pm$ 18 | 81 $\pm$ 6 |

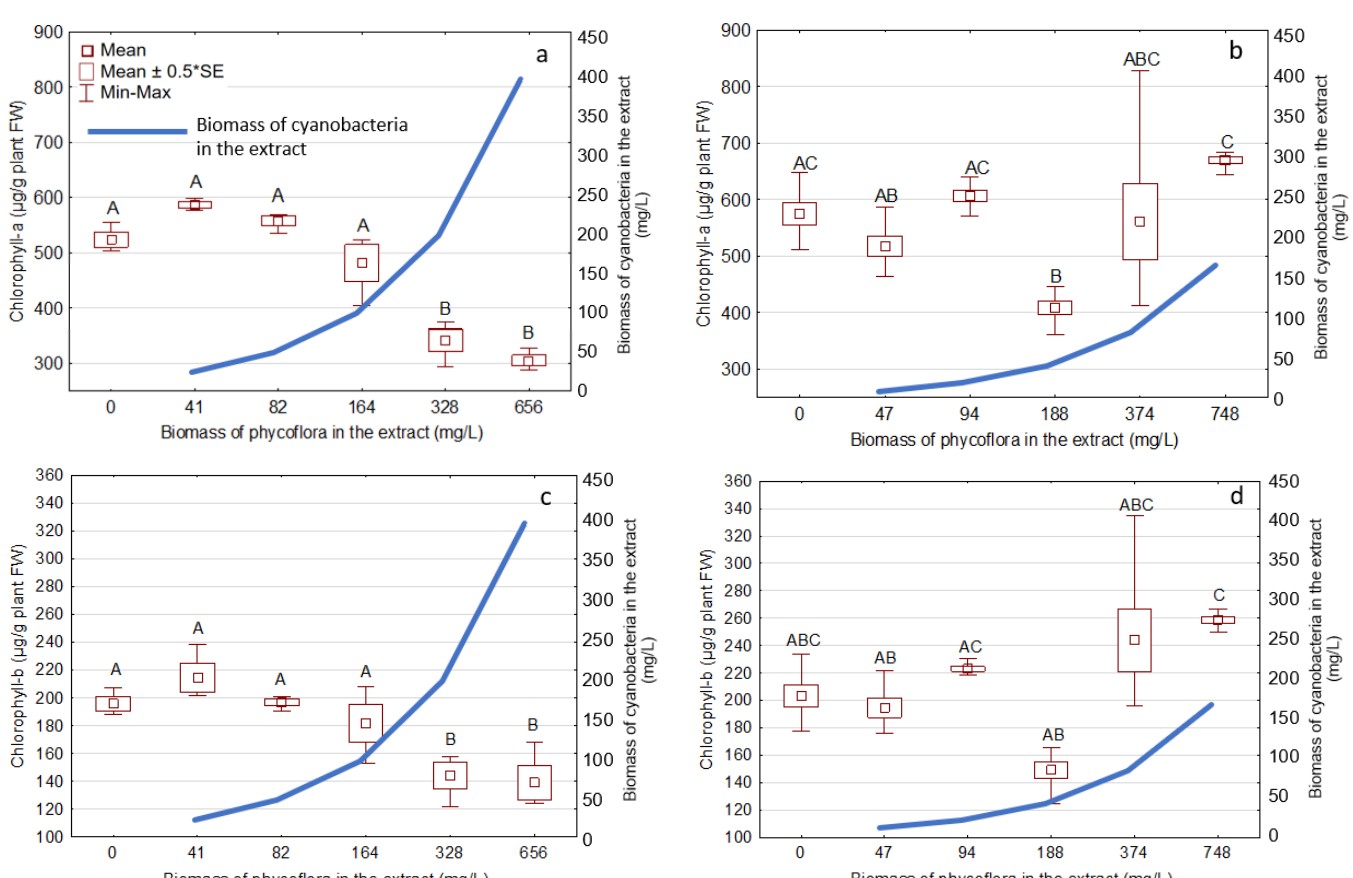

**Figure 4.** *Cont.*

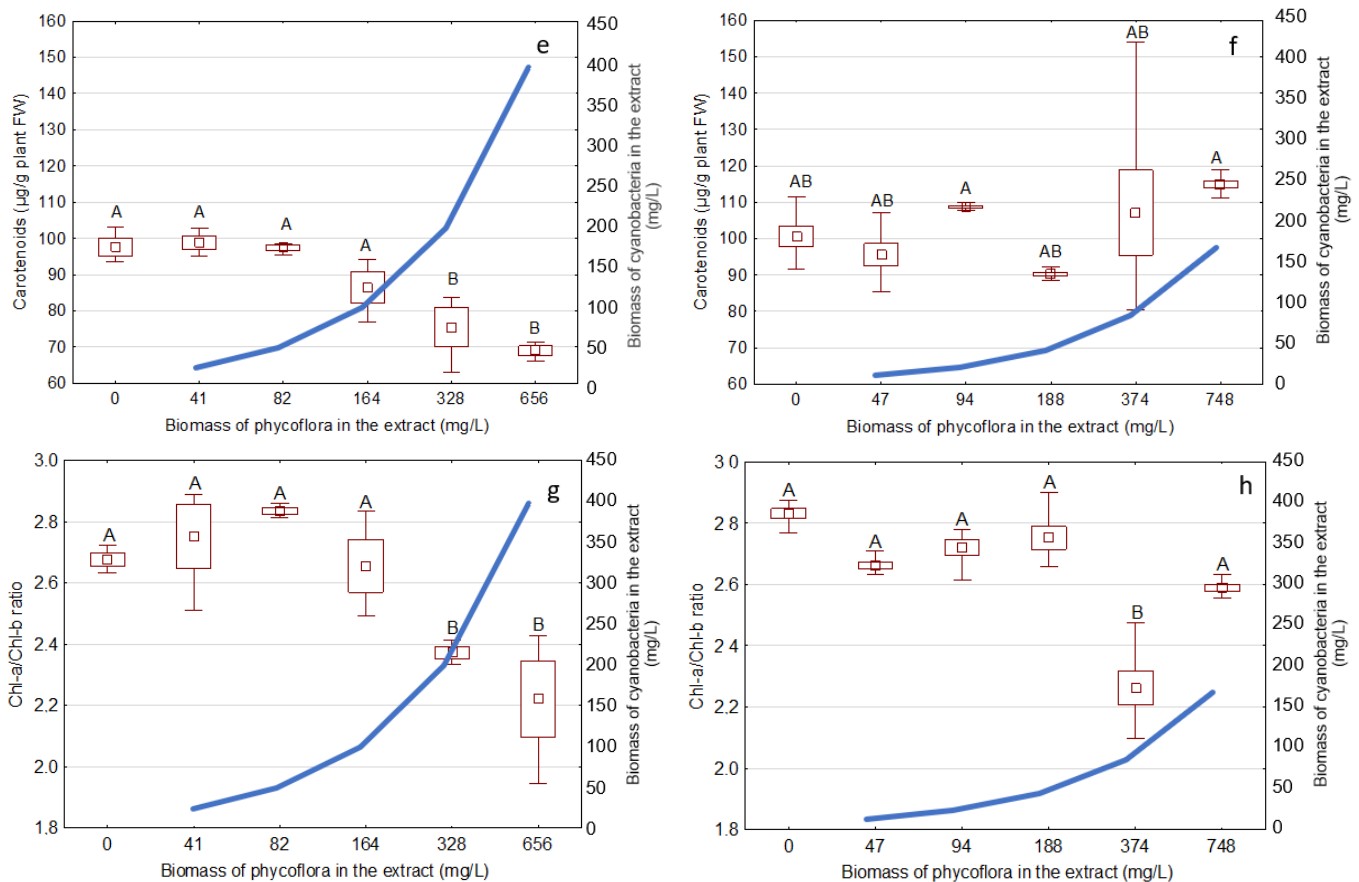

**Figure 4.** Content of Chlorophyll-a (Chl-a; (**a**,**b**)), Chl-b (**c**,**d**), carotenoids (Car, (**e**,**f**)), and the ratio of Chl-a to Chl-b (**g**,**h**) in *S. polyrhiza* exposed for 72 h to *Phormidium* extract (**a**,**c**,**e**,**g**) and *Oscillatoria* extract (**b**,**d**,**f**,**h**) (means ± SE, *n* = 3, 0 = controls). Different uppercase letters (A, B, C) indicate statistically significant differences between treatments (ANOVA, Tukey's test, $p < 0.05$).

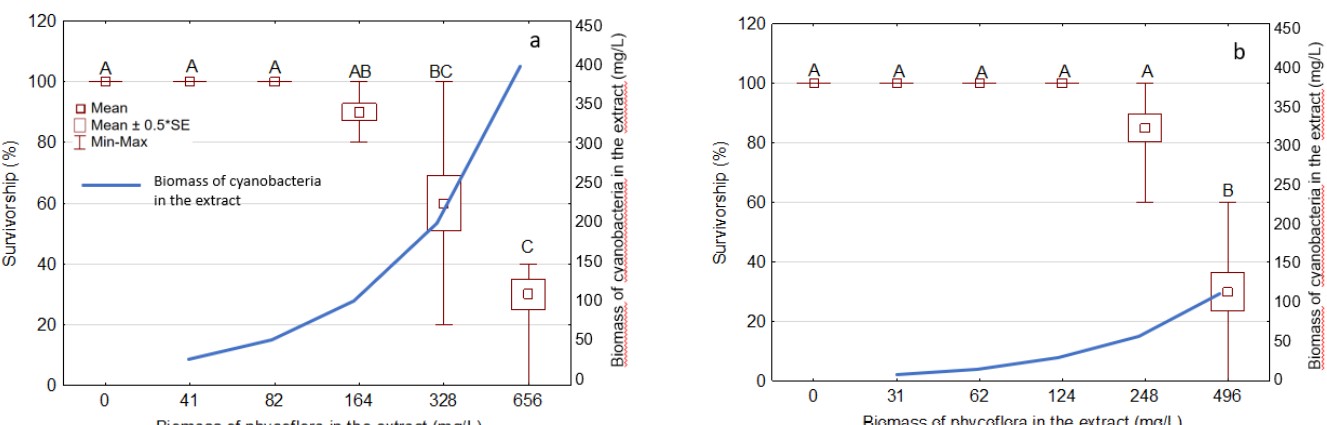

**Figure 5.** The influence of the *Phormidium* extract (**a**) and the *Oscillatoria* extract (**b**) on the survivorship of *D. magna* after 48 h exposure (mean ± SE; *n* = 4, 0 = controls). The survivorship of organisms in controls was set as 100%. Uppercase letters (A, B, C) point to statistically significant differences between treatments (ANOVA, Tukey's test, $p < 0.05$).

## 4. Discussion

This is the first report on the negative effects of secondary metabolites produced by epilithic and benthic phycoflora containing cyanobacteria from an ambient spring on the duckweed *S. polyrhiza* and zooplankters *D. magna*. In general, phycoflora (excluding diatoms) of ambient springs is very rarely studied [8] and referenced therein [9–11]. Our

results showed that the ambient limnokrenic Górecko spring is a heterogeneous habitat at the scale of a microhabitat, due to the difference in the quantitative and qualitative structure of phycoflora found for two studied microbial mats that were collected in the spring at the same time. This confirms that springs are microhabitat mosaics, and spring niches are unique habitats [41,42]. In the current study, four cyanobacterial taxa in each microbial mat were found, and the total number of cyanobacterial taxa was five. This is in agreement with previous studies of ambient springs [8]. The survey by Cantonati and co-authors [8] showed that the number of cyanobacterial taxa per spring varied between 0 and 10, with an average of four. We showed that cyanobacteria *O. limosa* and *P. breve*, commonly occurring in different aquatic ecosystems worldwide [6], were a very important part of microbial benthic mats (22 and 60% of the total phycoflora biomass, respectively) in the anthropogenically transformed spring, which confirmed some previous rare studies on cyanobacteria inhabiting ambient springs [8,10]. In the present study, among cyanobacteria-creating mats, the predomination (95%) of one filamentous species belonging to Oscillatoriales was observed. However, the domination of filamentous species in ambient springs appears to prevail in warm and arid climates, as in ambient springs in a temperate climate coccoid taxa predominate [8]. Nowicka-Krawczyk and Żelazna-Wieczorek [10] showed recently that among the environmental factors that have the greatest influence on cyanobacterial communities in ambient springs is a group of ions not related to natural conditions in the spring environment ($NO_2^-$, $NO_3^-$, $NH_4^+$, and $PO_4^{3-}$). The elevated levels (in comparison to other springs in the Roztocze region) of $NO_3$ and $PO_4$ were observed in the studied Górecko spring, and, for example, *O. limosa* is the species that prefers eutrophicated waters [33]. The presence of the above-mentioned ions in groundwater results from both direct and indirect human activity in the area of aquifers. Nowicka-Krawczyk and Żelazna-Wieczorek [10] concluded that the dynamics in cyanobacterial communities in the studied springs were stimulated by human impact and weather conditions.

Despite the massive occurrence of epilithic and benthic cyanobacteria in the studied Górecko spring, neither the production of microcystins (MCs) nor anatoxin-a (ANTX) was observed in the cyanobacterial biomass during the current study. However, since several of the species for which MC production was shown (*Schizothrix fasciculata*, *Rivularia* spp., and *Tolypothrix distorta*) [6,18–20] occur also in springs, it is very probable that cyanotoxins may be produced in ambient springs as well. Also, filamentous cyanobacteria belonging to Oscillatoriales that were present in the studied Górecko spring belong to potential cyanotoxin producers, and cyanotoxin production by benthic species is more and more often reported [6,43–46]. It was shown that some *Oscillatoria* and *Phormidium* strains may produce neurotoxic ANTX [6,43]. For instance, the benthic *Oscillatoria* sp. strain PCC 6506 was able to produce ANTX thanks to the presence of *anaA-H* genes and was a cylindrospermopsin (CYN) producer as well [43,44]. Gugger and co-workers [45] reported in a French river that the benthic cyanobacterium *Kamptonema* (*Phormidium*) *favosum* was producing ANTX, which was connected with dog neurotoxicosis. The concentration of ANTX reached from 1.8 to 15.3 µg ANTX/g of lyophilised weight in *Phormidium* biofilms. Benthic mat-forming cyanobacteria also occur commonly in New Zealand rivers frequently populated by *Phormidium*, known to produce ANTX and homoanatoxin (HTX), the latter at contents up to 4400 µg/g DW [46]. Izaguirre and co-authors [47] isolated from four reservoirs in southern California benthic cyanobacteria (Oscillatoriales) that produced microcystin-LR. Studies on cyanotoxin production in ambient springs are extremely rare [14,15]. For instance, recently [14] a new nodularin-producing benthic cyanobacterium, *Iningainema pulvinus* gen nov., sp. nov., was isolated in Australia from a freshwater ambient spring wetland. Two extracts from the strain contained 796 and 1096 µg of nodularin/g dry weight. On the other hand, *P. breve,* which occurred in Górecko spring, was isolated from benthos in a Brazilian pond [48] and did not contain mcyE or sxtA genes and did not produce MCs or ANTX.

To the best of our knowledge, this is the first report on the toxicity of metabolites produced by phycoflora, including cyanobacteria, in ambient springs. Although we did not

observe the production of known cyanotoxins, our study showed the negative effects of two tested crude extracts obtained from phycoflora biomasses containing 60% of *P. breve* and 22% of *O. limosa,* respectively, on duckweeds *S. polyrhiza* and daphnid *D. magna*. This suggests that cyanobacteria inhabiting springs may be harmful to aquatic organisms, particularly those living in or near microbial mats. Anderson and co-authors [49] showed that three *Phormidium* strains isolated from the Russian River (CA, USA) contained dihydroanatoxin-a and were highly toxic to macroinvertebrates *Hyalella azteca*, *Ceriodaphnia dubia*, and *Chironomus dilutes*. This confirms our results that *Phormidium* has the potential to impact macroinvertebrates, which play an essential role in aquatic environments. Macrophyte communities also play an important role in the functioning of freshwater ecosystems [50]. The duckweed *S. polyrhiza* occurs commonly in the temperate climatic zone and is a very good test organism [28,51]. We revealed, in the bioassays on *S. polyrhiza,* the negative effects of the extracts obtained from phycoflora inhabiting the ambient spring on the development of young plants and the production of pigments (chlorophylls and carotenoids). As it was a pioneer study carried out on biological material containing cyanobacteria collected from a spring habitat, we compared the results with those obtained on planktonic cyanobacteria. Pawlik-Skowrońska and co-authors [51] showed lastly the toxic effects of the crude extracts (Pa-A and Pa-B) obtained from two bloom samples predominated (95–97%) by oligopeptide-producing populations of the planktonic cyanobacterium *Planktothrix agardhii* on *S. polyrhiza*. The toxicity observed was dependent on the content of oligopeptides other than MCs. The 72 h bioassays showed a decrease in plant biomass by 35–63%, which was a higher inhibition than those observed in the present study (23–47%). Interestingly, the maximum biomasses of toxic *P. agardhii* in the extracts were ca. 115 and 443 mg/L, which is comparable to the cyanobacterial biomass concentration used in the current study. Moreover, in the present study, the number of roots decreased by 51–69% in comparison with controls in experiments with *Phormidium* extract and by 37–58% in the bioassays with *Oscillatoria* extract. These values are higher than those in experiments with planktonic oligopeptide-producing *P. agardhii* and suggest that the tested extracts contained harmful substances other than cyanotoxins. Our study also confirms that the root system of the duckweed is more sensitive to harmful compounds than the frond system.

We also observed the negative effect of the phycoflora extracts on pigment production of chlorophylls and carotenoids. Interestingly, Pawlik-Skowrońska and co-authors [51] did not observe a statistically important decrease in the Chl-a production by *S. polyrhiza* after exposure to the Pa-A extract of MC-producing *P. agardhii*; however, some statistically important effects were observed under the exposure to another extract (Pa-B) obtained from a different *P. agardhii* population which produced more oligopeptides other than MCs. Again, interestingly, concentrations of MCs seemed to be unimportant in the observed effects. Chlorophyll, the most abundant dye in plants, plays a key role in the biosynthesis processes occurring in green parts of plants. In the photosynthesis process, chlorophyll enables the conversion of light energy into the energy of chemical bonds. Chl-a is the main photosynthetic pigment and Chl-b is a ubiquitous accessory pigment, whereas the ratio of Chl-a to Chl-b controls the absorbed light intensity [32]. In higher plants, the Chl-a/Chl-b ratio is approximately 3:1 [52,53], and this value decreased under exposure to the tested extracts, showing a negative effect (stress) of the metabolites present in the extracts on photosystem II of the tested young plants *S. polyrhiza*. There are several factors that potentially complicate the relationship between Chl content and Chl-a fluorescence intensity [54]. In general, the Chl-a/Chl-b ratio has been predicted to decrease with the decrease in irradiance but responds differently under varying availability of nitrogen [55]. As shown recently [56], also at higher reactive oxygen species (ROS) concentrations, the content of chlorophyll decreases, which impairs the photosynthetic mechanism in plants. Our results clearly show that microbial mats containing cyanobacteria might produce substances other than MCs or ANTX that are harmful to aquatic macrophytes and their photosystems.

Our results also indicate that daphnids were sensitive to metabolites present in both extracts. The extract containing *O. limosa* was more toxic to *D. magna* than the extract

containing metabolites of *P. breve*. It was shown that *O. limosa* from a hot spring was able to produce MC-RR and MC-LR [22], whereas cyanotoxin production by *P. breve* was not proven [48]; however, studies in this field are scarce. To the best of our knowledge, there is no information about the ability of those two species to produce other cyanotoxins or known harmful compounds except for the taste/odor metabolite 2-methylisoborneol found in *O. limosa* [57]. The toxicity of cyanobacterial metabolites can be species-specific and may depend on the route of exposure–ingestion of cyanobacterial cells and/or absorption of biologically active dissolved compounds. For instance, as shown by Gilbert [58], the survivorship of the rotifer *Synchaeta pectinata* was affected at a five-times lower ANTX concentration (0.20 mg/L) than those (1 mg of ANTX/L) used in the experiment on *D. pulex,* at which survivorship did not decrease. Moreover, the day of first reproduction or interclutch interval was affected in a temperature-dependent manner. The effects that were observed in the current study might have been a consequence of physicochemical variation, different concentrations, and bioactivity of various metabolites present in the extracts. Experiments by Pawlik-Skowrońska et al. [17] confirmed much higher toxicity of the cyanobacterial extracts obtained from different planktonic species and containing mixtures of various cyanotoxins and other metabolites other than of pure toxins MC-LR and ANTX that were used in equivalent concentrations. The effect of planktonic cyanobacterial extracts rich in natural mixtures of several cyanobacterial metabolites on zooplankton has been extensively studied [59–63]. A few of these research studies characterized extracts for metabolites other than cyanotoxins. There is evidence that complex cyanobacterial bloom samples frequently exert adverse effects that cannot result from the presence of known cyanotoxins alone. In our experiments, the toxicity of the extracts to *D. magna* increased in a dose-dependent manner.

Planktonic cyanobacteria besides MCs may produce aeruginosins (AERs), microviridins (MDN D-F, J), cyanopeptolins, and cyclic anabaenopeptins (APs) [64–66]; however, nothing is known about the production of these substances by cyanobacteria inhabiting springs. Meanwhile, APs produced by several species of cyanobacteria are potent protease inhibitors [3]. In nature, they are as common as MCs [4,64–67]. The toxicity of cyanobacteria is determined not only by cyanotoxins but by the whole profile of bioactive compounds. The biological role of many cyanobacterial metabolites such as oligopeptides (over 700 structural variants) is still unclear and under discussion [4,17,67,68] and references therein). Our study shows that cyanobacteria and/or algae inhabiting springs may produce unknown secondary metabolites harmful to cladocerans. Méjean and co-authors [69] revealed that the benthic *Oscillatoria* strain, which produced ANTX and homoanatoxin-a, cytotoxic cylindrospermopsin, and two neurotoxins, contained other clusters of genes containing nonribosomal peptide synthase and polyketide synthase, likely involved in the biosynthesis of not-yet-identified secondary metabolites.

It cannot be excluded that metabolites of algae present in the extracts could play a role in the effects observed. For example, a recent report [62] showed that the extract of a common freshwater green alga, *Desmodesmus quadricauda*, was as cytotoxic and genotoxic as extracts of popular bloom-forming cyanobacteria, probably due to the production of retinoic acids (RAs). In the last decade, environmental retinoids, especially RAs, have been seen as an increasing concern because of their teratogenic effects and potential role in the development of deformations in aquatic vertebrates [70]. A recent survey by Gärtner and co-authors [21] shows that cyanobacteria are the biggest source of toxic substances in different, and also extreme, environments, but some algae may produce allelopathic and/or toxic compounds as well. It was shown that 47 genera of cyanobacteria, five of dinoflagellates, and one from each of the groups of green algae, golden algae, and diatoms can be related to phycotoxin and odor (T&O) compound production in different habitats. For instance, *Ulva lactuca* (Chlorophyta) producing hemolysins were responsible for differential toxic effects on the herbivorous gastropods, *Littorina littorea* and *L. obtusata* (Mollusca) [71], whereas one of the classical inhabitants of thermal springs, marine, and inland waters, the diatom *Amphora coffeaeformis*, is regarded as a producer of neurotoxic

domoic acid (DA) [72]. This toxin is produced by some marine diatoms and is harmful to fish, birds, marine mammals, and humans that live on contaminated fish. Therefore, we cannot exclude that the findings of the present study are a result of the simultaneous effects of cyanobacterial and algal metabolites.

## 5. Conclusions

Studying the cyanobacteria in ambient springs, which are endangered ecosystems, is important from the point of view of ecotoxicology, biodiversity, and conservation of water sources. We showed that mixtures of secondary metabolites of cyanobacteria and algae present in microbial benthic and epiphytic mats in ambient springs may cause toxic effects on macrophytes and zooplankters. Both tested extracts negatively affected *S. polyrhiza* development; however, they changed the pigment production in a different manner. The extract containing *O. limosa* metabolites was more toxic to *D. magna* than the extract containing higher amounts of metabolites of *P. breve*. Cyanobacteria and algae produce a vast diversity of bioactive substances that can be simultaneously present in aquatic environments, and further studies in this field are strongly required, especially in terms of looking for bioactive and potentially hazardous substances that may also be present in rarely studied ambient springs. The exact adaptive meaning of the potential production of cyanotoxins and other potentially harmful substances in springs has to be elucidated. It is very important, since waters from ambient springs are a source of water for river systems, to which potentially toxic substances released from cyanobacteria and algae inhabiting springs can travel.

**Author Contributions:** Conceptualization, M.T.; methodology, M.T., B.F. and J.D.; formal analysis, M.T., B.F. and J.D.; investigation, M.T., B.F. and J.D.; resources, M.T.; writing—original draft preparation, M.T.; writing—review and editing, B.F. and J.D.; validation, M.T.; visualization, M.T. and B.F.; supervision, M.T.; project administration, M.T.; funding acquisition, M.T. All authors have read and agreed to the published version of the manuscript.

**Funding:** This research was funded by the National Science Centre, Poland, grant no. 2019/03/X/NZ8/01442.

**Institutional Review Board Statement:** Not applicable.

**Informed Consent Statement:** Not applicable.

**Data Availability Statement:** Data are available upon request.

**Acknowledgments:** The author would like to thank the European Cooperation in Science and Technology, COST Action ES 1105 "CYANOCOST—Cyanobacterial blooms and toxins in water resources: Occurrence, impacts and management" for adding value to this study through networking and sharing knowledge with European experts and researchers in the field.

**Conflicts of Interest:** The authors declare no conflict of interest.

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
