# Peer review of "Effects of Extracts Containing Metabolites of Different Cyanobacteria from an Ambient Spring (Central Europe) on Zooplankters Daphnia magna and Duckweed Spirodela polyrhiza"

_water, doi:10.3390/w14244107_

Round 1

Reviewer 1 Report

Journal: Water (ISSN 2073-4441)

Manuscript ID: water-2046002

Type: Article

Title: Effects of crude extracts containing metabolites of different cyanobacteria from a temperate ambient spring (Eastern Poland) on zooplankters Daphnia magna and duckweed Spirodela polyrhiza

Section: Biodiversity and Functionality of Aquatic Ecosystems

Special Issue: Aquatic Ecology and Biological Invasions

This manuscript aimed to study the production of microcystins (MCs) and anatoxin-a by cyanobacteria in benthic microbial mats of Górecko spring, and the effects of the crude extracts on zooplankters Daphnia magna and the macrophyte Spirodela polyrhiza. However, there are many issues need to be addressed. The presentation is not good. I have the following comments and suggestions for the authors to improve the quality of manuscript.

1. Abstract

Lines 13-17

“Therefore, we studied the toxin content and effects of two extracts obtained from epilithic and benthic microbial mats containing different algae and filamentous cyanobacteria Phormidium breve and Oscillatoria limosa, respectively, on fresh biomass, a number of roots, and pigments content in duckweed Spirodela polyrhiza and on survivorship of Daphnia magna (Cladocera).”

Please change “a number of roots” to “number of roots”.

2. Line 23

“Two the highest concentrations”

Please check it.

3. Section “1. Introduction”

Lines 56-60

“For instance, some planktonic cyanobacteria can produce hepatotoxic microcystins (MCs) and other biologically active, mostly non-ribosomal oligopeptides such as aeruginosins, cyanopeptolins, anabaenopeptins, microviridins [3,15] as well as neurotoxic anatoxins (ANTXs) or saxitoxins (STXs) and many others substances that may negatively affect aquatic organisms [6].”

MCs are not only hepatotoxic, but can also cause nephrotoxicity, neurotoxicity, cardiovascular disease, immunomodulation, endocrine disruption, reproductive and developmental toxicity. Please See and cite the following paper.

Challenges of using blooms of Microcystis spp. in animal feeds: A comprehensive review of nutritional, toxicological and microbial health evaluation. https://doi.org/10.1016/j.scitotenv.2020.142319

4. Lines 65-75

“More studies demonstrate the production of toxins by benthic cyanobacteria inhabiting streams [16-18]. For instance, Fetscher and co-authors [18] demonstrated the production of MCs and ANTX in California streams. Over 1200 wadeable stream segments were conducted throughout the state, and the study revealed a high occurrence of potentially toxigenic benthic cyanobacteria. Moreover, benthic MCs were detected in one-third of the sites, based primarily on one-time sampling. The mean concentration of MCs was 46 μg/m2 of stream-bottom. Sites, where MCs were produced, spanned a variety of surrounding land-use types, from undeveloped land to heavily urbanized or agricultural places. Lyngbyatoxin, STXs, and ANTX were also detected, however, at lower concentrations than MCs. Aboal and co-workers [17] demonstrated that toxin production by cyanobacteria is widespread in high-ecological-integrity calcareous Mediterranean streams.”

What are the similarities and differences between springs and streams? Why did you introduce research of streams here?

5. Section “2. Materials and Methods”

Sub-section “2.1. Study area, field study and physical-chemical analysis of spring water”

What are the area and depth of the Górecko spring? Please insert the information in the revised manuscript.

6. Lines 127-129

“Collections were made in August 2020 at two sites, located in different parts of the Górecko spring, where biofilms were visually detectable as green or yellow-green microbial mats on stones and sediments.”

Please present the two sampling sites in the map of the Górecko spring.

7. Lines 137-138

“Biomass of cyanobacteria and algae was calculated according to Hillebrand and co-authots [28].”

Please change “co-authots” to “co-authors”. Also, the reference [28] is not by Hillebrand and co-authors. Please check the reference list.

8. Lines 164-165

“ANTX in extracts was analysed using liquid chromatography (HPLC) with fluorescence detection (Shimadzu, Kyoto, Japan) according to James and co-authors [29].”

The reference [29] is not by James and co-authors. Please check the reference list.

9. Line 186

“2.5.1.2 Analyses of growth and development parameters”

Line 192

“2.5.1.2.1. Analysis of pigments”

Line 198

“2.5.2 Acute toxicity bioassays on Daphnia magna”

Please check the number sequence of the titles.

10. Section “3. Results”

Lines 230-231

“The water temperature varied slightly and has not exceeded 10°C, which classifies the spring as ambient.”

Please present data of water temperature in the revised manuscript.

11. Lines 231-233

“Both pH value (7.38) and oxygen concentration (3.38 mg/L) showed the strong influence of groundwater on the physical-chemical conditions of spring water.”

How did you get the conclusion of “the strong influence of groundwater on the physical-chemical conditions of spring water”? Please add the information in the revised manuscript.

12. Lines 233-234

“Although the spring niche was affected by human activity, the water type does not evolve from HCO3Ca.”

How did you get the conclusion? Please add the information in the revised manuscript.

13. Lines 234-235

“The value of electrical conductivity (488 μS/cm) is within the typical range of Roztocze’s springs.”

What is the typical range of Roztocze’s springs? Please add the information in the revised manuscript.

14. Lines 235-237

“Only slightly elevated concentrations of anthropogenic K+ and SO42+ (2.6 and 25.2 mg/L, respectively) and nutrients (29.9 mg NO3/L and 0.41 mg PO4/L) were observed.”

Why did you say slightly elevated concentrations? To which data was compared? Please clearly describe it in the revised manuscript.

15. Lines 251-252

“HPLC analysis of intracellular cyanotoxins revealed neither MCs nor ANTX in microbial mats containing cyanobacteria (Table 1).”

Do Phormidium breve and Oscillatoria limosa only produce MCs and ANTX? Not other cyanotoxins? Why did you only measure MCs and ANTX?

16. Lines 259-263

“Both, after exposure to Phormidium extract and the Oscillatoria extract the average biomass of S. polyrhiza decreased almost twice (from 5.42 in the controls to 3.07 mg of FW in the treatment with the highest extract concentration used, and from 5.78 in the controls to 3.08 mg of FW in the penultimate concentration used, respectively; Figure 2).”

Please change “almost twice” to “43%”, (5.42-3.07)/5.42=43.4%

The value in the table 2 is right.

17. Lines 263-266

“The average number of roots (Figure 3) incubated with Phormidium extract decreased more than three times in comparison with the controls (from 6.5 to 2), whereas in the treatments with Oscillatoria extract, the parameter decreased two times (from 7.4 to 3.1).”

Please change “three times” to “70%”, (6.5-2)/6.5=69.2%.

Please change “two times” to “60%”, (7.4-3.1)/7.4=58.1%

The values in the table 2 are right.

18. Lines 268-270

“The strongest, most negative, and statistically significant (p<0.05) effect was observed at the highest extract concentrations used, particularly in the case of Phormidium extract.”

Which parameter do you mean? Please insert citation of Fig. 2 or Fig. 3.

19. Lines 298-300

“Chl-a content was promoted at two the lowest concentrations of Phormidium extract (41 and 82 mg/L) compare to the controls, whereas an increased Chl-b content was observed only at the lowest concentration used.”

In figure 4, no significant changes were observed.

20. Lines 298-301

“Chl-a content was promoted at two the lowest concentrations of Phormidium extract (41 and 82 mg/L) compare to the controls, whereas an increased Chl-b content was observed only at the lowest concentration used. However, the differences were not statistically significant (p>0.05; Figures 4 a,c).”

It is meaningless. There were no significant changes. Please delete the sentences. Please also check the whole manuscript.

21. Lines 327-330

“Since the biomass of cyanobacteria present in the extracts was 3.6 times higher in the Phormidium extract than in the Oscillatoria extract, the 48-h LC50 value based on cyanobacterial biomass was over 2-fold higher for Oscillatoria extract (81 mg/L) than for Phormidium extract (181 mg/L) (Table 3).”

Please change “higher” to “lower”.

22. Section “4. Discussion”

The 5th paragraph

Lines 449-505

This paragraph is too long. Please divide it to two paragraphs or more.

Author Response

Reviewer 1

This manuscript aimed to study the production of microcystins (MCs) and anatoxin-a by cyanobacteria in benthic microbial mats of Górecko spring, and the effects of the crude extracts on zooplankters Daphnia magna and the macrophyte Spirodela polyrhiza. However, there are many issues need to be addressed. The presentation is not good. I have the following comments and suggestions for the authors to improve the quality of manuscript.

 Response: Thank you very much for all your valuable suggestions. All were taken into consideration.

  1. Abstract

Lines 13-17

“Therefore, we studied the toxin content and effects of two extracts obtained from epilithic and benthic microbial mats containing different algae and filamentous cyanobacteria Phormidium breve and Oscillatoria limosa, respectively, on fresh biomass, a number of roots, and pigments content in duckweed Spirodela polyrhiza and on survivorship of Daphnia magna (Cladocera).”

Please change “a number of roots” to “number of roots”.

Response: It has been changed (line 16)

  1. Line 23

“Two the highest concentrations”

Please check it.

Response: The sentence describing the results was corrected and wrong information was deleted (line 24).

  1. Section “1. Introduction”

Lines 56-60

“For instance, some planktonic cyanobacteria can produce hepatotoxic microcystins (MCs) and other biologically active, mostly non-ribosomal oligopeptides such as aeruginosins, cyanopeptolins, anabaenopeptins, microviridins [3,15] as well as neurotoxic anatoxins (ANTXs) or saxitoxins (STXs) and many others substances that may negatively affect aquatic organisms [6].”

MCs are not only hepatotoxic, but can also cause nephrotoxicity, neurotoxicity, cardiovascular disease, immunomodulation, endocrine disruption, reproductive and developmental toxicity. Please See and cite the following paper.

Challenges of using blooms of Microcystis spp. in animal feeds: A comprehensive review of nutritional, toxicological and microbial health evaluation. https://doi.org/10.1016/j.scitotenv.2020.142319

Response: The information was included and the paper was cited (lines 62-64).

  1. Lines 65-75

“More studies demonstrate the production of toxins by benthic cyanobacteria inhabiting streams [16-18]. For instance, Fetscher and co-authors [18] demonstrated the production of MCs and ANTX in California streams. Over 1200 wadeable stream segments were conducted throughout the state, and the study revealed a high occurrence of potentially toxigenic benthic cyanobacteria. Moreover, benthic MCs were detected in one-third of the sites, based primarily on one-time sampling. The mean concentration of MCs was 46 μg/m2 of stream-bottom. Sites, where MCs were produced, spanned a variety of surrounding land-use types, from undeveloped land to heavily urbanized or agricultural places. Lyngbyatoxin, STXs, and ANTX were also detected, however, at lower concentrations than MCs. Aboal and co-workers [17] demonstrated that toxin production by cyanobacteria is widespread in high-ecological-integrity calcareous Mediterranean streams.”

What are the similarities and differences between springs and streams? Why did you introduce research of streams here?

 Response: Rivers in the Roztocze region, originate from the springs. Water physical-chemical parameters in the upper sections of rivers show similarity with springs. However, the paragraph is too long and it has been changed in the revised paper (lines 74-86).

  1. Section “2. Materials and Methods”

Sub-section “2.1. Study area, field study and physical-chemical analysis of spring water”

What are the area and depth of the Górecko spring? Please insert the information in the revised manuscript.

 Response: The information has been included in the revised paper (lines 140-141).

  1. Lines 127-129

“Collections were made in August 2020 at two sites, located in different parts of the Górecko spring, where biofilms were visually detectable as green or yellow-green microbial mats on stones and sediments.”

Please present the two sampling sites in the map of the Górecko spring.

 Response: Now sampling points are presented in the map of the Górecko spring (Fig. 1).

  1. Lines 137-138

“Biomass of cyanobacteria and algae was calculated according to Hillebrand and co-authots [28].”

Please change “co-authots” to “co-authors”. Also, the reference [28] is not by Hillebrand and co-authors. Please check the reference list.

Response: The corrections were made, and the reference number was corrected too (line 166).

  1. Lines 164-165

“ANTX in extracts was analysed using liquid chromatography (HPLC) with fluorescence detection (Shimadzu, Kyoto, Japan) according to James and co-authors [29].”

The reference [29] is not by James and co-authors. Please check the reference list.

Response: The reference list was checked and the reference numbers were corrected (lines 673-678).

  1. Line 186

“2.5.1.2 Analyses of growth and development parameters”

Line 192

“2.5.1.2.1. Analysis of pigments”

Line 198

“2.5.2 Acute toxicity bioassays on Daphnia magna”

Please check the number sequence of the titles.

Response: The number sequence of the titles was checked and corrected (lines 218-230).

  1. Section “3. Results”

Lines 230-231

“The water temperature varied slightly and has not exceeded 10°C, which classifies the spring as ambient.”

Please present data of water temperature in the revised manuscript.

 Response: The data are included in the revised paper (lines 264).

 Lines 231-233

“Both pH value (7.38) and oxygen concentration (3.38 mg/L) showed the strong influence of groundwater on the physical-chemical conditions of spring water.”

How did you get the conclusion of “the strong influence of groundwater on the physical-chemical conditions of spring water”? Please add the information in the revised manuscript.

 Response: This sentence was rather unfortunate (it should be corrected during proofreading). It was re-written in the resubmitted manuscript (lines 265-267).

  1. Lines 233-234

“Although the spring niche was affected by human activity, the water type does not evolve from HCO3Ca.”

How did you get the conclusion? Please add the information in the revised manuscript.

 Response: The conclusion was based on the ionic composition of water and this information was added in the revised manuscript (line 268).

  1. Lines 234-235

“The value of electrical conductivity (488 μS/cm) is within the typical range of Roztocze’s springs.”

What is the typical range of Roztocze’s springs? Please add the information in the revised manuscript.

 Response: The mean value of EC in Roztocze’s springs, being a result of multi-annual monitoring (from the 80.) of Chmiel (2001) amounted to 483 μS/cm. This information is provided in the revised paper (line 270).

  1. Lines 235-237

“Only slightly elevated concentrations of anthropogenic K+ and SO42+ (2.6 and 25.2 mg/L, respectively) and nutrients (29.9 mg NO3/L and 0.41 mg PO4/L) were observed.”

Why did you say slightly elevated concentrations? To which data was compared? Please clearly describe it in the revised manuscript.

Response: The concentration of parameters in question was elevated in relation to other springs. This was described in the revised paper, and the relevant citation was provided (lines 272-273). 

  1. Lines 251-252

“HPLC analysis of intracellular cyanotoxins revealed neither MCs nor ANTX in microbial mats containing cyanobacteria (Table 1).”

Do Phormidium breve and Oscillatoria limosa only produce MCs and ANTX? Not other cyanotoxins? Why did you only measure MCs and ANTX?

Response: Microcystins and anatoxin-a were analysed because these toxins are representatives of hepatotoxins and neurotoxins and are the most commonly produced by cyanobacteria (also benthic species). This explanation was added in the revised manuscript in the Introduction (line 117) and was discussed in the Discussion (lines 436-438, 501-505).

  1. Lines 259-263

“Both, after exposure to Phormidium extract and the Oscillatoria extract the average biomass of S. polyrhiza decreased almost twice (from 5.42 in the controls to 3.07 mg of FW in the treatment with the highest extract concentration used, and from 5.78 in the controls to 3.08 mg of FW in the penultimate concentration used, respectively; Figure 2).”

Please change “almost twice” to “43%”, (5.42-3.07)/5.42=43.4%

The value in the table 2 is right.

Response: It was changed (lines 297, 299), Table 2 was cited.

  1. Lines 263-266

“The average number of roots (Figure 3) incubated with Phormidium extract decreased more than three times in comparison with the controls (from 6.5 to 2), whereas in the treatments with Oscillatoria extract, the parameter decreased two times (from 7.4 to 3.1).”

Please change “three times” to “70%”, (6.5-2)/6.5=69.2%.

Please change “two times” to “60%”, (7.4-3.1)/7.4=58.1%

The values in the table 2 are right.

Response: It was changed (lines 301, 302).

  1. Lines 268-270

“The strongest, most negative, and statistically significant (p<0.05) effect was observed at the highest extract concentrations used, particularly in the case of Phormidium extract.”

Which parameter do you mean? Please insert citation of Fig. 2 or Fig. 3.

Response: The sentence was clarified and the citation of appropriate figures was added (lines 306, 308).

  1. Lines 298-300

“Chl-a content was promoted at two the lowest concentrations of Phormidium extract (41 and 82 mg/L) compare to the controls, whereas an increased Chl-b content was observed only at the lowest concentration used.”

In figure 4, no significant changes were observed.

Response: This sentence was deleted (lines 303-305).

  1. Lines 298-301

“Chl-a content was promoted at two the lowest concentrations of Phormidium extract (41 and 82 mg/L) compare to the controls, whereas an increased Chl-b content was observed only at the lowest concentration used. However, the differences were not statistically significant (p>0.05; Figures 4 a,c).”

It is meaningless. There were no significant changes. Please delete the sentences. Please also check the whole manuscript.

Response: This is true. This sentence was deleted (lines 303-305) and the whole manuscript was checked and similar sentences were deleted (lines 337-340, 349-351).

  1. Lines 327-330

“Since the biomass of cyanobacteria present in the extracts was 3.6 times higher in the Phormidium extract than in the Oscillatoria extract, the 48-h LC50 value based on cyanobacterial biomass was over 2-fold higher for Oscillatoria extract (81 mg/L) than for Phormidium extract (181 mg/L) (Table 3).”

Please change “higher” to “lower”.

Response: It was changed (line 368).

  1. Section “4. Discussion”

The 5th paragraph

Lines 449-505

This paragraph is too long. Please divide it into two paragraphs or more.

Response: The paragraph was divided into three paragraphs.

Reviewer 2 Report

Toporowska, Effects of crude extracts containing metabolites of different cyanobacteria from a temperate ambient spring (Eastern Poland) on zooplankters Daphnia magna and duckweed Spirodela polyrhiza

Rationale: cyanobacterial toxicity in ambient springs is unexplored. Substances that may negatively affect aquatic organisms are studied. Toxin content and effects of extracts from microbial mats containing filamentous cyanobacteria Phormidium breve and Oscillatoria limosa, on fresh biomass, a number of roots, and pigments content in duckweed Spirodela polyrhiza and on the survivorship of Daphnia magna (Cladocera). Aim and hypothesis: at the anthropogenically transformed Górecko spring, these extracts negatively affect D. magna survivorship and development of S. polyrhiza due to the presence of MCs and/ or ANTX.  

General: the title is too large and too local

Arguments for this study are at the end. To get the attention of a more general public it should start saying that the duck weed S. polyrhiza is a good model for measuring stress, also refer to the turion and root importances (The same is valid for D. magna). Duckweed relevance appears only in line 404.

The species Spirodela polyrhiza (giant duckweed) is a cosmopolite representative of the Lemnoideae subfamily in standardized ecotoxicological test procedures (Environment Canada 2007) S. polyrhiza is also widely applied as a model organism in plant physiology, ecotoxiciology and bioremediation studies.

It is also worth explaining the reason why of the output variables like Chla, Chlb, carotenes: Chl-a is the main photosynthetic pigment, whereas Chl-b is a ubiquitous accessory pigment in plants, and the Chl-a/Chl-b content ratio controls the absorbed light intensity, etc.

Limnokren type of ambient spring, needs to be explained.

I think it is very interesting the finding abut the microbial mat composition… worth expanding. Why is it relevant that in both mats, three accompanying cyanobacterial taxa occurred and that their biomass did not exceed 5% of the total cyanobacterial biomass.

Specific comments:

Lines 331  Phorimdium should be Phormidium

Lines 371-373 there is a mistake

In the Oscillatoria 314 extract results are strange, in Phormidium Biomass 328 also, the standard error is too large, this should be explained if it could be an artifact.

Author Response

Reviewer 4

Response: Thank you very much for your opinion and valuable suggestions. All were taken into consideration.

Comments and Suggestions for Authors

Toporowska, Effects of crude extracts containing metabolites of different cyanobacteria

from a temperate ambient spring (Eastern Poland) on zooplankters Daphnia magna and

duckweed Spirodela polyrhiza

Rationale: cyanobacterial toxicity in ambient springs is unexplored. Substances that may

negatively affect aquatic organisms are studied. Toxin content and effects of extracts from

microbial mats containing filamentous cyanobacteria Phormidium breve and Oscillatoria

limosa, on fresh biomass, a number of roots, and pigments content in duckweed Spirodela

polyrhiza and on the survivorship of Daphnia magna (Cladocera). Aim and hypothesis: at the

anthropogenically transformed Górecko spring, these extracts negatively affect D. magna

survivorship and development of S. polyrhiza due to the presence of MCs and/ or ANTX.

General: the title is too large and too local

Response: The title was shortened and changed.

Arguments for this study are at the end. To get the attention of a more general public it should

start saying that the duck weed S. polyrhiza is a good model for measuring stress, also refer

to the turion and root importances (The same is valid for D. magna). Duckweed relevance

appears only in line 404.

The species Spirodela polyrhiza (giant duckweed) is a cosmopolite representative of the

Lemnoideae subfamily in standardized ecotoxicological test procedures (Environment

Canada 2007) S. polyrhiza is also widely applied as a model organism in plant physiology,

ecotoxiciology and bioremediation studies.

Response: The information about the role of bioassays using D. magna and S. polyrhiza was added (lines 110-114) and new references were included (lines 653-659).

It is also worth explaining the reason why of the output variables like Chla, Chlb, carotenes:

Chl-a is the main photosynthetic pigment, whereas Chl-b is a ubiquitous accessory pigment

in plants, and the Chl-a/Chl-b content ratio controls the absorbed light intensity, etc.

Response: The information was added (lines 122-125).

Limnokren type of ambient spring, needs to be explained

 Response: The term was confusing and it was checked and explained correctly (lines 41-42 and 108).

I think it is very interesting the finding about the microbial mat composition… worth

expanding. Why is it relevant that in both mats, three accompanying cyanobacterial taxa

occurred and that their biomass did not exceed 5% of the total cyanobacterial biomass.

Response: Thank you for your opinion. It was important due to the structure of microbial mats and cyanobacterial communities observed (predomination of one cyanobacterial species lines 392-400).

Specific comments:

Lines 331 Phorimdium should be Phormidium

Response: Corrected (line 371).

Lines 371-373 there is a mistake

Response: The information was corrected (lines 390-391)

In the Oscillatoria 314 extract results are strange, in Phormidium Biomass 328 also, the

standard error is too large, this should be explained if it could be an artifact.

Response: All results were checked carefully during experiments, interpretation of the results, manuscript preparation and now, during revision. Generally, the standard errors are larger in experiments with Oscillatoria extract (particularly at concentration 374 in the case of pigments) than in those with Phormidium extract. However, at the concentration 374 mg/L (Oscillatoria extract) results for biomass and number of roots had the lowest values in comparison with other treatments and SD was the lowest (Figs. 2b and 3b). Plants from those experiments were used to analyse pigments (the results varied widely). Removing the extreme scores distorts the final results, therefore, we ask for agreement to live the presentation of the results in a present form.

Reviewer 3 Report

The paper of Toporowska et al. presented the effects of crude extracts containing metabolites of different cyanobacteria from a temperate ambient spring (Eastern Poland) 3 on zooplankters Daphnia magna and duckweed Spirodela polyrhiza.

The presentation and quality of the paper are commendable. However, I have a few minor suggestions for the author.  The introduction part of the paper seems to lack information about the role of bioassays such as the D. magna and S. polyrhiza. I highly appreciate it if the authors would cite papers that have previously used these bioassays for ecotoxic studies. I recommend these papers:

1. Reyes, V. P., Ventura, M. A., & Amarillo, P. B. (2022). Ecotoxicological Assessment of Water and Sediment in Areas of Taal Lake with Heavy Aquaculture Practices Using Allium cepa and Daphnia magna Assay. Philippine Journal of Science, 151(3), 969–974.

2. Pietrini, F., Iannilli, V., Passatore, L., Carloni, S., Sciacca, G., Cerasa, M., & Zacchini, M. (2022). Ecotoxicological and genotoxic effects of dimethyl phthalate (DMP) on Lemna minor L. and Spirodela polyrhiza (L.) Schleid. Plants under a short-term laboratory assay. Science of The Total Environment, 806, 150972. https://doi.org/10.1016/j.scitotenv.2021.150972

3. Connors, K. A., Brill, J. L., Norberg‐King, T., Barron, M. G., Carr, G., & Belanger, S. E. (2022). Daphnia magna and Ceriodaphnia dubia Have Similar Sensitivity in Standard Acute and Chronic Toxicity Tests. Environmental Toxicology and Chemistry, 41(1), 134–147. https://doi.org/10.1002/etc.5249

     Author Response

Reviewer 3

The paper of Toporowska et al. presented the effects of crude extracts containing metabolites of different cyanobacteria from a temperate ambient spring (Eastern Poland) 3 on zooplankters Daphnia magna and duckweed Spirodela polyrhiza.

The presentation and quality of the paper are commendable. However, I have a few minor suggestions for the author.  The introduction part of the paper seems to lack information about the role of bioassays such as the D. magna and S. polyrhiza. I highly appreciate it if the authors would cite papers that have previously used these bioassays for ecotoxic studies. I recommend these papers:

1. Reyes, V. P., Ventura, M. A., & Amarillo, P. B. (2022). Ecotoxicological Assessment of Water and Sediment in Areas of Taal Lake with Heavy Aquaculture Practices Using Allium cepa and Daphnia magna Assay. Philippine Journal of Science151(3), 969–974.

2. Pietrini, F., Iannilli, V., Passatore, L., Carloni, S., Sciacca, G., Cerasa, M., & Zacchini, M. (2022). Ecotoxicological and genotoxic effects of dimethyl phthalate (DMP) on Lemna minor L. and Spirodela polyrhiza (L.) Schleid. Plants under a short-term laboratory assay. Science of The Total Environment806, 150972. https://doi.org/10.1016/j.scitotenv.2021.150972

  1. Connors, K. A., Brill, J. L., Norberg‐King, T., Barron, M. G., Carr, G., & Belanger, S. E. (2022). Daphnia magnaand Ceriodaphnia dubia Have Similar Sensitivity in Standard Acute and Chronic Toxicity Tests. Environmental Toxicology and Chemistry41(1), 134–147. https://doi.org/10.1002/etc.5249

Response: Thank you very much for your opinion and valuable suggestions. All were taken into consideration. The information about the role of bioassays using D. magna and S. polyrhiza was added (lines 110-116) and references were included (lines 657-662).

Reviewer 4 Report

The title show be revised and shortened

Abstract did not cover effect of these toxic metabolites on Daphnia, the zooplankton sample

Abstract: Line 23, the word “Two” is it indicating the second point, then it should have a comma afterward, moreover, there is no “One” to indicate a listing.

Correct sentences in Line 48 and Line 55

The topic gave the impression that you are investing the effect of metabolites from the extracts, which may me both known and unknown, but your methodology focused on two, MCs and ANTX. If focusing on this two, then just indicate it in the title.

From the result it seems MCs and ANTX was not detected, then the methodology should have been redesigned to eliminate the analysis of MCs and ANTX, and the author would gave tried to identify the major metabolites present, since the extracts still should some negative effect even with the absence of MCs and ANTX.

The plant used for the bioassay text was refer to as “old plant”, does this mean a matured plant? And wouldn’t this after the growth pattern on its own, which may mask the negative effect from exposure to metabolites in the extracts?

Author Response

Reviewer 4

Comments and Suggestions for Authors

Response: Thank you very much for your opinion and valuable suggestions. All were taken into consideration.

The title show be revised and shortened.

Response: The title was revised and shortened.

Abstract did not cover effect of these toxic metabolites on Daphnia, the zooplankton sample

Response: The information was added (lines 27-28).

Abstract: Line 23, the word “Two” is it indicating the second point, then it should have a comma afterward, moreover, there is no “One” to indicate a listing.

Response: “Two” does not indicate the second point, but the number of the highest concentrations of Phormidium extract that had a statistically significant effect on S. polyrhiza. We ask for permission of living this sentence at its present form. 

Correct sentences in Line 48 and Line 55

Response: The sentences were corrected (lines 52 and 60).

The topic gave the impression that you are investing the effect of metabolites from the extracts, which may me both known and unknown, but your methodology focused on two, MCs and ANTX. If focusing on this two, then just indicate it in the title.

Response: Yes, that is true, we investigated the effects of unknown metabolites (mixture of metabolites). The methodology focuses on nine MC variants and anatoxin-a as they are the most commonly produced by cyanobacteria. We highlighted it in the introduction (lines 117, 127). Therefore,  we ask for leaving the present title.

From the result it seems MCs and ANTX was not detected, then the methodology should have been redesigned to eliminate the analysis of MCs and ANTX, and the author would gave tried to identify the major metabolites present, since the extracts still should some negative effect even with the absence of MCs and ANTX.

Response: We plan to collect more cyanobacterial samples and study the production of oligopeptides other than MCs. We ask for permission of living the description of methods as they are important (we show which MC standards were used to look for particular MC, other unidentified could be present in the extracts as well)

The plant used for the bioassay text was refer to as “old plant”, does this mean a matured plant? And wouldn’t this after the growth pattern on its own, which may mask the negative effect from exposure to metabolites in the extracts?

Response: In line 207, there is “72-h-old plant” which means that the plants were 3 days old at the beginning of the experiment. They were young which was added in line 215 and stated in the discussion in lines 295, 456 and  489.

Round 2

Reviewer 1 Report

Journal: Water (ISSN 2073-4441)

Manuscript ID: water-2046002-peer-review-v2

Type: Article

Title: Effects of extracts containing metabolites of different cyanobacteria from an ambient spring (Central Europe) on zooplankters Daphnia magna and duckweed Spirodela polyrhiza

Section: Biodiversity and Functionality of Aquatic Ecosystems

Special Issue: Aquatic Ecology and Biological Invasions

This manuscript aimed to study the production of microcystins (MCs) and anatoxin-a by cyanobacteria in benthic microbial mats of Górecko spring, and the effects of the extracts on zooplankters Daphnia magna and the macrophyte Spirodela polyrhiza.

The manuscript improved during the revisions. However, there are still some issues to be addressed. I have the following comments and suggestions for the authors to improve the quality of manuscript.

1. Abstract

The location of the spring should be described in the abstract.

2. Line 23

“Two the highest concentrations”

What do you mean? Please rephrase it.

3. Section “2. Materials and Methods”

Lines 161-162

“Diatoms were identified mostly up to the genus level according to Cox [32].”

The reference [32] is not by Cox. Please check the reference list. I have made some comments on the reference list last time, but authors only revised the errors I pointed out. Please check the entire reference list.

4. Line 224

“2.5.1.2.1. Analysis of pigments”

Please check the number sequence of the titles.

5. Section “3. Results”

Lines 267-269

“Although the spring niche was af- 267

fected by human activity, the water type of wate, (determined by ionic composition) does not evolve from HCO3Ca.”

Please present data of ionic composition in the revised manuscript.

6. Lines 296, 300-301

Please change “decreased to” to “decreased by”. Please check the data and the descriptions in the entire manuscript. “decreased to” and “decreased by” are different.

7. Please read and carefully check through all the manuscript text, tables, figures and supplementary materials. It is the authors’ responsibility to present their best work to the readers.

Author Response

Responses to the Reviewer

  1. Abstract

The location of the spring should be described in the abstract.

Response: The location of the spring was presented in the Abstract (lines 18-19).

  1. Line 23

“Two the highest concentrations”

What do you mean? Please rephrase it.

Response: The “Two” in question was an unnecessary residue after the sentence correction. It has been deleted (line 25). 

  1. Section “2. Materials and Methods”

Lines 161-162

“Diatoms were identified mostly up to the genus level according to Cox [32].”

The reference [32] is not by Cox. Please check the reference list. I have made some comments on the reference list last time, but authors only revised the errors I pointed out. Please check the entire reference list.

Response: Thank you for this comment. Yes, there was a mistake which was corrected (line 165). All the references were checked carefully.

  1. Line 224

“2.5.1.2.1. Analysis of pigments”

Please check the number sequence of the titles.

Response: The numbers have been corrected and are now in a continuous manner (lines 221-233).

  1. Section “3. Results”

Lines 267-269

“Although the spring niche was af- 267

fected by human activity, the water type of water, (determined by ionic composition) does not evolve from HCO3Ca.”

Please present data of ionic composition in the revised manuscript.

Response: The data of ionic composition has been presented (lines 272-275).

  1. Lines 296, 300-301

Please change “decreased to” to “decreased by”. Please check the data and the descriptions in the entire manuscript. “decreased to” and “decreased by” are different.

Response: Thank you for the remark. The issue has been addressed within the manuscript (lines 302, 307).

  1. Please read and carefully check through all the manuscript text, tables, figures and supplementary materials. It is the authors’ responsibility to present their best work to the readers.

Response: The paper has been double-checked. Few errors were found and corrected (mostly unnecessary words and typos, lines 72, 241, 349, 440, 569, Table 1)   

Reviewer 2 Report

Suggested changes have been incorporated, also important improvements.

Author Response

Thank you very much for your positive opinion. 

Reviewer 4 Report

ok

Author Response

(The authors gave the same response as above.)
